# Structured tailored rehabilitation after hip fragility fracture: The 'Stratify' feasibility and pilot randomised controlled trial protocol

Katie J. Sheehan[1,2]*, Stefanny Guerra[1], Salma Ayis[2], Aicha Goubar[2], Nadine E. Foster[3], Finbarr C. Martin[2], Emma Godfrey[2], Ian D. Cameron[4], Celia L. Gregson[5], Nicola E. Walsh[6], Anna Ferguson Montague[7], Rebecca Edwards[8], Jodie Adams[8], Gareth D. Jones[8], Jamie Gibson[8], Catherine Sackley[9], Julie Whitney[2]

1 Bone & Joint Health, Blizard Institute, Queen Mary University of London, London, United Kingdom, 2 Department of Population Health Sciences, School of Life Course and Population Health, Faculty of Life Science and Medicine, King's College London, London, United Kingdom, 3 Surgical Treatment and Rehabilitation Service (STARS) Education and Research Alliance, The University of Queensland and Metro North Health, Queensland, Australia, 4 John Walsh Centre for Rehabilitation Research, Northern Sydney Local Health District and University of Sydney, Sydney, Australia, 5 Musculoskeletal Research Unit, Translation Health Sciences, Bristol Medical School, University of Bristol, Bristol, United Kingdom, 6 Centre for Health and Clinical Research, University of the West of England Bristol, Bristol, United Kingdom, 7 Public and Patient Involvement Member Representative from Trauma Rehabilitation (Orthopaedic) for Older People (TROOP), London, United Kingdom, 8 Guy's and St Thomas's NHS Foundation Trust, London, United Kingdom, 9 School of Health Sciences, Queens Medical Centre, University of Nottingham, Nottingham, United Kingdom

* k.sheehan@qmul.ac.uk, katie.sheehan@kcl.ac.uk

**Data Availability Statement:** No datasets were generated or analysed during the current study. All relevant data from this study will be made available upon study completion.

## Abstract

### Background

Rehabilitation in hospital is effective in reducing mortality after hip fracture. However, there is uncertainty over optimal in-hospital rehabilitation treatment ingredients, and the generalizability of trial findings to subgroups of patients systematically excluded from previous trials. The aim of this study is to determine the feasibility of a randomized controlled trial which aims to assess the clinical- and cost-effectiveness of adding a stratified care intervention to usual care designed to improve outcomes of acute rehabilitation for all older adults after hip fracture.

### Methods

This is a protocol for a single site, feasibility and pilot, pragmatic, parallel group (allocation ratio 1:1) randomised controlled assessor-blind STRATIFY trial (**S**tructured **T**ailored **R**ehabilitation **A**f**T**er **HI**p **F**ragilit**Y** Fracture). Adults aged 60 years or more, surgically treated for hip fracture following low energy trauma (fragility fracture), who are willing to provide consent or by consultee declaration (depending on capacity), are eligible. Individuals who experienced in-hospital hip fracture will be excluded. Screening, consent/assent, baseline assessment (demographics, patient reported outcome measures or PROMs [health related quality of life, activities of daily living, pain, falls related self-efficacy], and resource use), and randomization will take place within the first four days post-admission. Participants will then

**Funding:** This work was supported by a United Kingdom Research and Innovation (UKRI) Future Leaders Fellowship in the form of a grant [MR/S032819/1] to KJS.

**Competing interests:** I have read the journal's policy and the authors of this manuscript have the following competing interests: KS received a grant from UK Research & Innovation Future Leaders Fellowship to support this work. This funding provides salary support for KS. KS also received funding from the National Institutes of Health Research (NIHR) and the Royal Osteoporosis Society for hip fracture health services research. KS is the Chair and CG a member of the Scientific and Publications Committee of the Falls and Fragility Fracture Audit Programme which managed the National Hip Fracture Database audit at the Royal College of Physicians. FCM was the funded (2012-2018) board chair of the Falls and Fragility Fracture programme. NEF is funded through an Australian National Health and Medical Research Council (NHMRC) Investigator Grant (ID: 2018182). CS and NW receive funding from the National Institute for Health Research (NIHR). CS is an NIHR Senior Investigator. Salma Ayis was funded/supported by the National Institute for Health Research (NIHR) Biomedical Research Centre based at Guy's and St Thomas' NHS Foundation Trust and King's College London. The views expressed are those of the author(s) and not necessarily those of the NHS, the NIHR or the Department of Health. CLG receives funding from Versus Arthritis (ref 22086). GSdP, SA, and IDC have no competing interests to declare. This does not alter our adherence to PLOS ONE policies on sharing data and materials.

be offered usual care, or usual care plus STRATIFY intervention during their hospital stay. The STRATIFY intervention includes 1) a web-based algorithm to allocate participants to low- medium- or high-risk subgroups; and 2) matched interventions depending on subgroup allocation. The low-risk subgroup will be offered a self-management review, training in advocacy, and a self-managed exercise programme with support for progression, in addition to usual care (1-hour 40 minutes therapist time above usual care). The medium-risk subgroup will be offered education, a goal-orientated mobility programme (with carer training, as available and following carer consent), and early enhanced discharge planning, in addition to usual care (estimated 2-hours 15 minutes therapist time above usual care). The high-risk subgroup will be offered education, enhanced assessment, orientation, and a goal-orientated activities of daily living programme (with carer training, as available and following carer consent), in addition to usual care (estimated 2-hours 45minutes therapist time above usual care). All STRATIFY subgroup treatment interventions are specified using the Rehabilitation Treatment Specification System (RTSS) for treatment theory development and replication. Follow-up PROM data collection, RESOURCE USE alongside readmissions and mortality, will be collected on discharge and 12-weeks post-randomisation. Intervention acceptability will be determined by semi-structured interviews with participants, carers, and therapists at the end of the intervention.

## Dissemination

The trial findings will be disseminated to patients and the public, health professionals and researchers through publications, presentations and social media channels.

## Trial registration

The trial has been registered at clinicaltrials.gov (NCT06014554).

## Introduction

Each year, United Kingdom (UK) hospitals admit 70,000 men and women over the age of 60 years with hip fracture [1]. Even with surgery, 30% of patients die within a year [2]. Among survivors, 25% never walk again, and 22% transition from independent living to nursing homes [2]. This led 81 global societies to endorse a call to action to improve acute multidisciplinary care after hip fracture [3].

A recent Cochrane systematic review supports rehabilitation in hospital as an effective approach to reduce mortality and adverse outcomes after hip fracture [4]. However, the nature of the rehabilitation interventions varied considerably limiting conclusions on the optimal treatment ingredients. This uncertainty has translated to NICE guideline recommendations being limited to daily mobilization and regular physiotherapy review [5]. In turn, international audit of rehabilitation after hip fracture is limited to an indicator of physiotherapy review and/or early mobilization [6].

The generalizability of the evidence is also limited as many rehabilitation trials attempted to account for differences in the hip fracture population by targeting homogenous subgroups [7]. Alternatively, patient subgroups may be excluded based on low perceived potential for benefit, low feasibility of participation or outcome data collection, or outside the target population [8].

This has led to the systematic exclusion of 27.3% potential participants based on equity-related factors from previous trials of rehabilitation after hip fracture, with nursing home residency and cognitive impairment the main drivers of these exclusions [8]. It is therefore uncertain whether interventions deemed 'effective' are so for all patients, or for targeted subgroups only.

This poses challenges as to how these interventions may be implemented when finite healthcare resources are allocated. Indeed, resources are often assigned where the evidence of what is best practice is clear and compelling and may lead to local prioritization [9]. In the absence of evidence, services for patients systematically excluded from trials may be under-prioritized leading to a potentially unfair distribution of therapeutic benefits.

Stratified rehabilitation may provide important solutions. Stratified rehabilitation identifies population subgroups with different risk of poor outcomes [10]. Subgroups are then matched to rehabilitation better tailored to their needs in an effort to improve outcomes for all patients alongside more equitable allocation of resources. Hip fracture survivors recently described this tailored approach as key to successful recovery [11]. Although an approach of matching rehabilitation to patient subgroups with different risks of poor outcomes is intuitive, it has not been tested for rehabilitation after hip fracture.

## Aims and objectives

The overarching purpose of a future definitive trial is to determine the clinical- and cost-effectiveness of adding a stratified care intervention to usual care designed to improve outcomes of acute rehabilitation for older adults after hip fracture. Several uncertainties will first be addressed through a randomized feasibility and pilot trial which this protocol details.

The primary objective of this feasibility and pilot randomized trial is to determine the treatment fidelity of the proposed intervention.

Secondary objectives seek to determine:

1. Count of screened, eligible, approached, recruited, and retained participants and carers.

2. The acceptability of the intervention to participants, carers, and physiotherapists/occupational therapists and/or therapy assistants.

3. Barriers and enablers to intervention delivery.

4. Acceptability, completeness, and descriptive comparison of outcome data collection.

5. Count of inadvertent unblinding of outcome assessors.

6. Count of adverse events (AE) and serious adverse events (SAE).

7. Indicative sample size for a future definitive trial.

## Methods

The protocol is reported in accordance with the Standard Protocol Items: Recommendations for Interventional Trials (SPIRIT) Checklist [12] (S1–S5 Files) and registered at clinicaltrials.gov (NCT06014554). The trial has received approval from East of England–Cambridge Central Research Ethics Committee and the Health Research Authority (REF: 24/EE/0005).

### Intervention

Stratified rehabilitation requires 1) the identification of subgroups with different risk of poor outcomes, and 2) rehabilitation tailored to the needs of each subgroup [10].

**Subgrouping tool.**   A subgrouping tool 'the stratify hip algorithm' was developed and validated (based on three multivariable prediction models) to identify patients at low-, intermediate- and high-risk of death and/or change in residence (to a higher level of care) using records for over 170,000 patients admitted to one of 173 hospitals in England and Wales. The description of the tool development and validation is available elsewhere [13]. The approach requires entry of 5 pieces of information to a website (age, sex, prefracture mobility, prefracture residence, and dementia diagnosis) to subgroup each patient [13]. A backup version is also available on an Excel spreadsheet. Overall, 31% of patients were assigned low-risk, 28% medium-risk, and 41% high-risk across three outcomes (in-hospital death, change in residence, 30-day death) [13]. The algorithm was developed for information/reference only. In this feasibility study the algorithm is intended to be used to stratify patient subgroups to matched rehabilitation intervention, according to the subgroup intervention protocol below.

**Subgroup interventions.**   Matched interventions were designed for each subgroup identified by the algorithm.

**Theoretical framework.**   Normalisation Process Theory [14] is a theoretical framework from implementation science which seeks to embed a practice into 'work as usual' through four components:

1. Coherence–understanding and making sense of a practice (here 'stratified approach to care').

2. Cognitive participation–engagement of physiotherapists and occupational therapists with the practice.

3. Collective action–the joint 'work' of physiotherapists, occupational therapists, patients (and carers) needed to enact the practice.

4. Reflexive monitoring–physiotherapist and occupational therapist reflection and appraisal of the practice over time to ensure it becomes routinely embedded.

This theory was selected to underpin development to optimise treatment fidelity of the subsequent matched intervention.

**Development.**   To inform the matched interventions, the research team completed qualitative interviews with patients [15], physiotherapists [16], and the multidisciplinary team [17], an umbrella review of systematic reviews of older adults who underwent rehabilitation interventions in the acute setting [18], and consultations with the public and patient involvement group 'TROOP' (further detail of TROOP available at www.ppitroop.co.uk). The findings were considered alongside recent systematic (including Cochrane) reviews [4, 19–22] and international guidelines [5, 23–25].

The intervention components (and their respective ingredients) were then determined by evidence-informed consensus using a Nominal Group Technique [26]. The Nominal Group Technique is a structured method to reach consensus in small group discussions conformed of a panel of experts on the topic of interest. The approach includes introduction, silent idea generation, round robin, clarifications, and scoring. It has been successfully modified for remote delivery [27]. For the current intervention, the panel was made up of 11 members: four physiotherapists, two clinical academic physiotherapists, two occupational therapists, one Professor of Rehabilitation, one patient representative with lived experience of hip fracture, and one carer representative with lived experience of supporting an older adult after orthopedic trauma.

An e-booklet was prepared and sent to panel members to review before attendance at a remote workshop (S6 File). To summarise, the e-booklet contained:

1. a context statement to set the scene in terms of the target population, where and when the intervention would take place, and target outcomes.

2. the approach taken to identify subgroups, along with their defining characteristics.

3. the questions to answer in the workshop: *For low-, medium-, and high- risk subgroups, what treatment components should be included in physiotherapy and occupational therapy management to optimize outcomes of inpatient rehabilitation*?

4. a summary of the Nominal Group Technique process.

5. a description of the theoretical framework to inform the interventions.

6. patient and carer perspective materials inclusive of a word cloud from a patient and public involvement focus group, summary of a qualitative synthesis on patient perspectives of recovery after hip fracture completed by the research team, and summary of a qualitative synthesis on carer perspectives of supporting recovery after hip fracture.

7. a summary of synthesized evidence from hip fracture guidelines, systematic reviews on hip fracture rehabilitation, older adults' rehabilitation and behavior change interventions in hospital settings.

At the workshop, the materials in the e-booklet were summarised in a presentation. Participants were then invited to generate ideas for the low-risk subgroup silently for 10-minutes (videos and sound off). During this time a set of potential intervention domains were displayed on the screen to stimulate ideas (Domains: exercise and activity, self-management, equipment, mood/anxiety, knowledge, discharge planning, engaging informal/formal carers). Participants were informed this was a suggested list and by no means exhaustive. Following silent idea generation, all participants were invited to contribute one idea at a time until all ideas were exhausted. Contributed ideas were input to a form (organised by suggested domains) visible to all via screen share in real time. After clarifications were sought and made, the process was repeated for the medium-risk and subsequently high-risk subgroups.

After the workshop, all ideas generated across all risk subgroups were converted into statements to proceed with the Scoring stage of the Nominal Group Technique. Two days later, an anonymous survey was circulated to workshop participants for anonymous scoring of statements on a 9-point Likert scale rating from 'Strongly disagree' to 'Strongly agree', or 'Not important' to 'Very important'. Participants responses were imported into an Excel spreadsheet and analysed for consensus. Consensus was determined when over 75% of participants 'Disagreed' or deemed a statement 'Not important' (scores 1–3), were 'Neutral' (scores 4–6), or 'Agreed' or deemed a statement as 'Important' (scores 7–9). Sensitivity of statements to outliers was explored.

For each risk subgroup, statements that reached consensus (consensus statements) were independently assigned by two researchers to constructs informed by the Taxonomy of Behaviour Change Techniques [28]. Constructs statements were subsequently developed to reflect all consensus statements within a given construct. Construct statements were then classified according to whether they described 'usual care' or not in consultation between the research team and a subgroup of panel members (n = 5). Conflicts were resolved by consensus.

**Intervention specification.**   Construct statements classified as 'new' were considered treatment ingredients, further detailed, and incorporated into the final matched interventions. The low-risk subgroup will be offered a self-management review, training in advocacy, and a self-managed exercise programme with support for progression, in addition to usual care (estimated 1-hour 40 minutes therapist time above usual care per participant). The medium-risk

subgroup will be offered education, a goal-orientated mobility programme (with carer training as available), and early enhanced discharge planning, in addition to usual care (estimated 2-hours 15 minutes therapist time above usual care). The high-risk subgroup will be offered education, enhanced assessment, orientation, and a goal-orientated activities of daily living programme (with carer training as available), in addition to usual care (estimated 2-hours 45 minutes therapist time above usual care per participant).

Active ingredients of rehabilitation interventions are rarely specified in research protocols which encapsulates typical criticisms of a 'black box' problem in the field [29, 30]. The Rehabilitation Treatment Specification System (RTSS) counteracts the problem because it provides a coherent framework based on treatment theory that compels rehabilitation professionals to specify active rehabilitative content (ingredients), that via theoretical processes (mechanisms of action), cause measurable change on desired function systems (targets) [31].

Attempts to resolve the black-box problem include development of the Template for Intervention Description and Replication (TIDieR) which is welcome guidance for describing the abundant interventions that fail to be conveyed by medication doses [32]. Abundancy in rehabilitation interventions is a mixed blessing. On one hand it relates to a thriving, diverse field. On the other it renders reporting guidance a challenge. The RTSS in contrast allows researchers to specify which items from the diverse descriptions of rehabilitation ingredients are reportable in experimental research protocols. This is because the RTSS is based on the concept of treatment theory and not solely on description, a distinction which might facilitate consensus on rehabilitation RCT guidance in the future [33]. Therefore, low, medium, and high-risk STRATIFY rehabilitation interventions were specified using the RTSS over 4 weeks by two senior physiotherapists (GDJ, JG) independent of the nominal group panel members who developed the interventions. The specifiers are members of the ACRM Treatment Specification Working Group [34] with more than 3 years experience of developing the RTSS in clinical practice [35–39]. They specified the interventions individually based on the RTSS handbook (v6.2) [40] and met three times to reach consensus on the final specifications. The RTSS specification was reviewed by the research team to ensure preservation of matched intervention detail generated from consensus construct statements. Full details of the proposed matched interventions are available in Tables 1–3.

The matched interventions will start before the fourth postoperative day and be delivered during the inpatient stay. Intervention components will be delivered by a physiotherapist, occupational therapist, or therapy assistant (hereafter referred to as 'therapist') depending on staffing availability. Treatment case report forms (CRFs) will be used to record the 'Stratify' intervention after each interaction with a therapist.

**Usual care.**   Patients randomized to the control arm will receive usual physiotherapy and occupational therapy care. Usual care entails physiotherapy and occupational therapy from the day after surgery to the point of discharge, with a focus on discharge planning and sufficient recovery of activities of daily living and mobility for safe return to prefracture residence. Treatment CRFs will be used to record usual care intervention after each interaction with a therapist.

## Design

This is a feasibility and pilot, pragmatic, single centre, two parallel arm, randomised controlled trial with an embedded qualitative study.

The primary objective of this study (to determine the treatment fidelity of the proposed intervention) will be met by analysis of data collected through written patient diaries, treatment CRFs (usual care intervention and Stratify intervention), and observations during Stratify intervention delivery.

**Table 1. STRATIFY Treatment Specification (RTSS)–Low-risk subgroups.**

| Description of clinical interaction and/or physical assessment dependencies | TARGET* What / In What Way | Group† | Volition Type‡ | Low Risk Group MOA§ | INGREDIENTS Ingredient‖ | Dosing Parameter / Progression |
|---|---|---|---|---|---|---|
| Assessment of barriers to self-management with the self-efficacy for managing chronic disease 6-item scale [41]. If a participant indicates a score ≤5 on an item, the therapist provides guidance within first 3 post-operative days to overcome identified barriers considering their capability, the opportunities, and their motivations [42] to maximise volitional performance of the self-managed exercise programme as directed | Performance of fatigue management strategies / as directed | R | V | Cognitive & Affective information processing | Therapist-led conversations in fatigue management including but not limited to: developing a propensity to exercise or be mobile in pursuit of their recovery in phase with higher predicted energy levels during the day, encourage pacing by dosing exercises and/or daily functions little-and-often, and encourage principles of utilising visitors to provide support and reassurance to practice exercise/mobility | n/a |
| | Performance of pain management strategies / as directed | R | V | Cognitive & Affective information processing | Therapist-led conversations in pain management including but not limited to: providing knowledge of the spectrum of pain management options the participant is entitled to and developing propensity to request or decline pain management options when needed or offered, and developing a propensity to align pain management to epochs of exercise or mobility practice in pursuit of participant's recovery | n/a |
| | Performance of emotional distress management strategies / as directed | R | V | Cognitive & Affective information processing | Therapist-led conversations in emotional distress management including but not limited to: providing knowledge of comparative social situations and the advantages of articulating concerns as reassurance; developing propensity and intentions to initiate dialogue with and request feedback from friends, family and healthcare providers to provide emotional support for ongoing recovery, provide knowledge to notify the direct care team to appropriate psychiatry or social care specialties for social welfare | n/a |
| | Performance of other symptom management strategies / as directed | R | V | Cognitive & Affective information processing | Therapist-led conversations on other symptoms management including but not limited to: providing knowledge of the advantages of articulating any other symptoms/health problem experiences to the direct care team and developing propensity and intention to approach the direct care team accordingly to prevent limiting ongoing recovery | n/a |
| | Performance of independent exercise and mobility for recovery / as directed | R | V | Cognitive & Affective information processing | Therapist/nurse/AHP-led conversations on independent exercise and mobility but not limited to: reassuring participants about their capability for independent exercise and mobility, providing knowledge about opportunities to safely perform supported independent exercise and mobilisation and its advantage in the pursuit of participant's recovery, developing opportunities to discuss concerns and solutions about independent exercise/mobility, develop the propensity to reflect on their exercise and mobility training with staff and visitors | n/a |
| Therapist trains advocacy: Outline participant's capability to complete activities of daily living, transfer, and mobilise independently during their hospital stay. Therapist encourages patients to do these activities for themselves at the representation level (restorative care approach [43]), even if staff or visitors offer to do them, as the activities are a part of their rehabilitation and recovery. Discuss how it can be easier to accept help, but the motivation for declining is to ensure their ongoing recovery. | Performance of restorative movement practice / as directed | R | V | Cognitive & Affective information processing | Therapist-led discussion in a coaching approach that includes knowledge about the link between benefits of performing activities of daily living and recovery, risks of learned non-use, and adaptation to activity avoidance. | n/a |
| The therapist provides opportunity for the patient to practice the self-managed exercise programme and is satisfied they can execute and progress it. | Ability to perform independent exercise programme / increase | S&H | DV | Learning by doing | Provide opportunity to practice exercise programme components, provide verbal feedback and/or demonstration on correct start positions, reciprocal movements, and progressions | Duration: 30 minutes |

(Continued)

**Table 1.** (Continued)

| | | | | | Low Risk Group | |
|---|---|---|---|---|---|---|
| **TARGET\*** | | | | | **INGREDIENTS** | |
| Description of clinical interaction and/or physical assessment dependencies | What / In What Way | Group† | Volition Type‡ | MOA§ | Ingredient\|\| | Dosing Parameter / Progression |
| Progressive exercise programme inclusive of resistance and endurance training tailored to functional baseline based on programme based on a published exercise training programme and as specified below [44], taught by a therapist who will populate a training plan template for the participant, which they can implement independently. | Knee & hip extension strength / increase | O | DV | Neuro-muscular modulation | Standing bilateral reciprocal squats | Frequency: 3<br>Sets: 3<br>Reps: 10<br>Rest: between sets [e.g. to resting level of acute fatigue].<br>Intensity: 20 degrees knee flexion<br>Progressions including, but not limited to: knee angle to 45 degrees, use sit-stand-sit from chair with extra cushion, progress to remove cushion with knee angle at ~120% lateral malleolus to knee joint line distance |
| | Hip abduction & extension strength / increase | O | DV | Neuro-muscular modulation | Standing unilateral hip abduction and reciprocal squats | Frequency: 3<br>Sets: 2<br>Reps: 10<br>Rest: between sets [e.g. to resting level of acute fatigue].<br>Intensity: Hip Abduction 30 degrees, knee flexion 20 degrees<br>Progressions including, but not limited to: Addition of proprietary resistance bands (e.g. attached to chair leg green (light) if able to complete 10reps without need for rest, progress through resistance (low-higher, yellow to red, then green, blue, black) |
| | Ankle plantarflexion strength / increase | O | DV | Neuro-muscular modulation | Standing on floor, bilateral heel raises to max available range | Frequency: 3<br>Sets: 3<br>Reps: 10<br>Rest: between sets [e.g. to resting level of acute fatigue]. Intensity: within max available range<br>Progressions including, but not limited to: difficulty to 2cm, then 4cm block under forefeet to increase dorsiflexion start position. Final progression, unilateral heel raise foot flat in floor 2x10 reps each side with rests as above |
| | Cardiovascular endurance / increase | O | DV | Cardio-vascular adaptation | Walking performed with assistance and mobility aid as required | Frequency: 3x total, 1x daily<br>Distance: 200m<br>Intensity: low intensity<br>Progression including, but not limited to: 600m at moderate intensity with aids as required. Increase frequency > 1x daily. |

\* Targets refer to patient unless otherwise stated [40]. Direct Target—change in the specific aspect of functioning which is predicted to result from performance of the treatment activity.

†Target groups—O (Organ functions), S&H (Skills and Habits), or R (Representations). Organ functions are treatments in which the functions of organs or organ systems are modified, often by systematic stimulation; Skills and Habits are treatments that have in common learning or improving a skill via practicing, or reducing the effort required/increasing the habitual nature of a behavioural routine; Representations are treatments aimed at changing internal (i.e., central nervous system) representations related to cognitions, affect, motivation, and intentions to perform volitional behaviours.

‡Volition Type—whether the change in the direct target is predicted to result from performance of the treatment activity by either the clinician (non-volitional treatments—NV), or the treatment recipient (volitional treatments—DV). The direct target for volitional treatments (DV) may be accompanied by a separately specified volition target (V) addressing the volitional behaviour required to achieve the direct target in cases where clinicians are unable to verify that the activity has been performed as prescribed to convey the active ingredients for the direct target.

§Mechanism of Action—the process by which a treatment's active ingredients induce change in the direct (or volition) target of treatment.

\|\|Ingredients—the observable, measurable and active elements of treatment that are hypothesised to directly change the target of treatment.

**Table 2. STRATIFY Treatment Specification (RTSS)–Medium-risk subgroups.**

| | | | | | | |
|---|---|---|---|---|---|---|
| **Medium Risk Group** | | | | | | |
| | **TARGET** | | | | **INGREDIENTS** | |
| Description of clinical interaction | What / In What Way (Measure) | Group* | Volition Type† | MOA# | Ingredient | Dosing Parameter |
| In the first week of participation the participant is provided an information pack detailing what to expect from rehabilitation and their recovery. The therapist discusses the content in the first week (paced and reinforced) with the participant and/or their carer. | Knowledge about rehabilitation and recovery expectations / increase | R | DV | Cognitive & Affective information processing | Written information pack & therapist-led discussion to participant and/or carer on: determining what getting better means, how rehabilitation supports recovery, identifying who is involved in the rehabilitation process, whether there are any changes to participants ability to think or remember clearly, understanding the likely pathway of care and realistic waiting times, understanding the extent of therapy available in the community and the role of charities to support recovery | n/a |
| | Carer knowledge about rehabilitation and recovery expectations / increase | | | | | |
| | Knowledge of pain management / increase | R | DV | Cognitive & affective information processing | Written information pack & therapist-led discussion on: understanding usual care processes for requesting pain management, determining intentions to request pain relief prior to activity, understanding barriers to this and alternatives that can be offered | n/a |
| | Knowledge of fear of falling / Reduce | R | DV | Cognitive & affective information processing | Written information document & therapist-led discussion on: reinforcing fear of falling is normal after a fall, encouraging activity as a positive driver of recovery, providing positive feedback on improvement of movement regularly. | n/a |
| | Performance of self-regulation of activity / as directed | R | V | Cognitive & affective information processing | Written information document & therapist-led discussion on: understanding the benefits of carrying out activities independently, understanding the rationale of reinforcing activity by exercising outside of supervised rehabilitation to support recovery. | n/a |
| | Propensity to engage in discharge planning / increase | R | DV | Cognitive & affective information processing | Written information document & therapist-led discussion on: understanding the advantages of co-created discharge planning and decisions, understanding if decisions have not been shared and if so reflecting on how to be involved in alternative decision making. | n/a |

(*Continued*)

**Table 2.** (Continued)

| Medium Risk Group | | | | | | |
|---|---|---|---|---|---|---|
| The therapist sets ambitious mobility programme goals together with participant ±carer (s) and subsequently performs skill practice to engender the capability of the participant and carer to complete the mobility skills training, whilst providing direct supervision as appropriated for mobility specific training. | Participant goal setting ability / increase<br><br>Carer goal setting ability / increase | S&H | DV | Learning by doing | Collaborative goal setting practise opportunities with therapist, participant and/or carer. Therapist description of difference between outcome goals and behaviour goals to achieve the outcome goal. Outcome goals set should seek to go beyond those set as part of usual care. Behaviour goals should incorporate early supported mobility aid progression and early incorporation of dual task mobility. | Duration: 10 minutes |
| | Mobility function / Increase | S&H | DV | Learning by doing | Supervised mobility training opportunities within context of mobility outcome and behaviour goals. e.g. if outcome goal is: I can walk outside on flat ground continuously for 3 minutes with the use of two crutches and accompanied by another person by the time I am leaving the hospital, the therapist will provide tailored mobility skill training ingredients conducive to initial behaviour goal e.g.: walk continuously indoors on flat ground for 3 minutes with the use of two crutches independently. | e.g.<br>Frequency: 5<br>Sets: 5<br>Duration: 3 minutes.<br>Progression: e.g. walk indoors around a set of obstacles on flat ground while using two crutches independently |
| | Propensity to perform physical rehabilitation / increase | R | DV | Cognitive & affective information processing | Completion of progress chart documentation, placing of progression chart at end of bed, completion of goal setting including documentation. | n/a. |
| | Carer ability to perform mobility training / increase | S&H | DV | Learning by doing | Therapist-led demonstration of practical skills required to support specified mobility training. Opportunity for carer to practise in presence of therapist, therapist verbal feedback and availability to answer carer questions | Duration: 30 minutes |
| | Carer knowledge of appropriate mobility involvement / Increase | R | DV | Cognitive & affective information processing | Therapist-led discussion of progress chart, Therapist led identification of when carer can practice mobility function practice and when not, identification of how to contact therapist if unsure or unconfident to support activity practice | n/a |

(*Continued*)

**Table 2.** (Continued)

| | | | | | | |
|---|---|---|---|---|---|---|
| **Medium Risk Group** | | | | | | |
| The therapist plans the discharge process with a discussion together with participant ±carer(s) to enable completion of participant-held rehabilitation discharge plan template which includes six questions and a multifactorial falls risk assessment** | Knowledge of community therapy availability / increase | R | DV | Cognitive & affective information processing | Therapist-led discussion and completion of discharge template of physiotherapy and occupational therapy, how to liaise with community professions, encouragement of early liaison, identification of unsuitability if patient not appropriate for service, identification of GP as contact source to arrange further support. | n/a |
| | Knowledge of appropriate independent/ carer led rehabilitation on discharge / increase | R | DV | Cognitive & affective information processing | Therapist-led discussion and completion of discharge template of current mobility goals which can be continued in community either independently or in presence of carer and how to safely progress rehabilitation. | n/a |
| | Knowledge of safe return to previous activity / increase | R | DV | Cognitive & affective information processing | Therapist-led discussion and completion of discharge template, patients previously completed activities, identification of safe activities to return to, guidance on how to return to approach return to activity. | n/a |
| | Knowledge of upcoming appointments/ increase | R | DV | Cognitive & affective information processing | Therapist-led discussion and completion of discharge template to identify likely waiting times and upcoming appointments. | n/a |
| | Knowledge of use important of documents / increase | R | DV | Cognitive & affective information processing | Therapist-led discussion and completion of multifactorial falls risk assessment and discharge plan, identify the benefit of showing to community therapist | n/a |
| | Knowledge on how to contact care professionals / increase | R | DV | Cognitive & affective information processing | Therapist-led discussion and completion of discharge template to identify who and how to contact community therapist and GP. | n/a |

* Targets refer to patient unless otherwise stated [40]. Direct Target—change in the specific aspect of functioning which is predicted to result from performance of the treatment activity.

†Target groups–O (Organ functions), S&H (Skills and Habits), or R (Representations). Organ functions are treatments in which the functions of organs or organ systems are modified, often by systematic stimulation; Skills and Habits are treatments that have in common learning or improving a skill via practicing, or reducing the effort required/increasing the habitual nature of a behavioural routine; Representations are treatments aimed at changing internal (i.e., central nervous system) representations related to cognitions, affect, motivation, and intentions to perform volitional behaviours.

‡Volition Type—whether the change in the direct target is predicted to result from performance of the treatment activity by either the clinician (non-volitional treatments—NV), or the treatment recipient (volitional treatments—DV). The direct target for volitional treatments (DV) may be accompanied by a separately specified volition target (V) addressing the volitional behaviour required to achieve the direct target in cases where clinicians are unable to verify that the activity has been performed as prescribed to convey the active ingredients for the direct target.

§Mechanism of Action–the process by which a treatment's active ingredients induce change in the direct (or volition) target of treatment.

||Ingredients–the observable, measurable and active elements of treatment that are hypothesised to directly change the target of treatment.

**Template with falls history, gait, balance, mobility, strength, perceived functional ability, concerns about falls, visual impairment, cognitive impairment, urinary incontinence, home hazards, postural hypotension, and polypharmacy), the date of assessment (this may vary across domains), how they were assessed, the result, and any recommendation for teams in the post-acute setting.

**Table 3. STRATIFY Treatment Specification (RTSS)–High-risk subgroups.**

| Description of clinical interaction and/or physical assessment dependencies | What / In What Way | Group* | Volition Type† | MOA# | Ingredient | Dosing Parameter |
|---|---|---|---|---|---|---|
| **High Risk Group** | | | | | | |
| | **TARGET** | | | | **INGREDIENTS** | |
| In the first week of participation the participant is provided an information pack detailing what to expect from rehabilitation and their recovery. The therapist discusses the content in the first week (paced and reinforced) with the participant and/or their carer. | Knowledge of rehabilitation process / increase | R | DV | Cognitive & affective information processing | Written information document & therapist-led discussion on: What does getting better mean to patient, how rehabilitation supports recovery, identification of who is involved in process. | n/a |
| | Knowledge of pain management / increase | R | DV | Cognitive & affective information processing | Written information pack & therapist-led discussion on: understanding usual care processes for requesting pain management, determining intentions to request pain relief prior to activity, understanding barriers to this and alternatives that can be offered. | n/a |
| | Knowledge of Fear of falling / reduce | R | DV | Cognitive & affective information processing | Written information document & therapist-led discussion on: reinforcing fear of falling is normal after a fall, encouraging activity as a positive driver of recovery, providing positive feedback on improvement of movement regularly. | n/a |
| | Performance of self-regulation of activity / as directed | R | DV | Cognitive & affective information processing | Written information document & therapist-led discussion on: benefits of carrying out activities independently, rationale of reinforcement of activity outside of supervised rehabilitation to support recovery, identification of limited therapy time. | n/a |
| Participants and/or carers (as available) requested by therapist to bring memoir materials for bedside. | Carer and patient knowledge of orientation risk and strategies / increase | R | DV | Cognitive & affective information processing | Therapist-led discussion on features of disorientation, whether there are current features of disorientation as well as the rationale and instruction of providing memoir materials (e.g. framed photographs of family and friends, photo album, music device, reading materials, pillowcase/blanket) to promote orientation. | n/a |
| Enhanced assessment and onward referral as required** | n/a (assessment based referral process) | – | – | – | – | |
| The therapist sets ADL programme goals together with participant ±carer(s) and subsequently performs skill practice to engender the capability of the participant and carer to complete the ADL skills training whilst providing direct supervision as appropriate for ADL specific training. | Participant goal setting ability / increase | S&H | DV | Learning by doing | Therapist description of difference between outcome and behaviour goals and their interrelationship. Collaborative goal setting practise opportunities with therapist, participant and/or carer. | Duration: 15 minutes |
| | Carer goal setting ability / increase | | | | | |

(*Continued*)

**Table 3.** (Continued)

| | TARGET | | | | INGREDIENTS | |
|---|---|---|---|---|---|---|
| | **High Risk Group** | | | | | |
| | ADL function / increase | S&H | DV | Learning by doing | Supervised ADL training opportunities within context of ADL outcome and behaviour goals. e.g. if outcome goal is: with verbal direction and provision of a soaped sponge, I can wash myself in a seated shower chair by the time I am leaving the hospital, the therapist will provide tailored ADL training ingredients conducive to initial behaviour goal e.g.: In a seated position, lift left thigh without use of hands. | Frequency: 3 Duration: 15minutes Progressions including but not limited to: addition of controlled lowering of thigh after lift, then progressive thigh lift holds from 1, to 3 and finally 5s. |
| | Propensity to perform physical rehabilitation / increase | R | DV | Cognitive & affective information processing | Completion of progress chart, placing of progression chart at end of bed, completion of goal setting. | n/a. |
| | Carer ability to perform ADL training / increase | S&H | DV | Learning by doing | Therapist-led demonstration of practical skills required to support specified ADL training. Opportunity for carer to practise in presence of therapist, therapist verbal feedback and availability to answer carer questions | Duration: 30 minutes |
| | Carer knowledge of appropriate ADL involvement / Increase | R | DV | Cognitive & affective information processing | Therapist-led discussion of progress chart, Therapist led identification of when carer can practice ADLs and when not, identification of how to contact therapist if unsure or unconfident to support activity practice | n/a |

ADL–activities of daily living' GP–general practitioner

* Targets refer to patient unless otherwise stated [40]. Direct Target—change in the specific aspect of functioning which is predicted to result from performance of the treatment activity.

†Target groups–O (Organ functions), S&H (Skills and Habits), or R (Representations). Organ functions are treatments in which the functions of organs or organ systems are modified, often by systematic stimulation; Skills and Habits are treatments that have in common learning or improving a skill via practicing, or reducing the effort required/increasing the habitual nature of a behavioural routine; Representations are treatments aimed at changing internal (i.e., central nervous system) representations related to cognitions, affect, motivation, and intentions to perform volitional behaviours.

‡Volition Type—whether the change in the direct target is predicted to result from performance of the treatment activity by either the clinician (non-volitional treatments—NV), or the treatment recipient (volitional treatments—DV). The direct target for volitional treatments (DV) may be accompanied by a separately specified volition target (V) addressing the volitional behaviour required to achieve the direct target in cases where clinicians are unable to verify that the activity has been performed as prescribed to convey the active ingredients for the direct target.

§Mechanism of Action–the process by which a treatment's active ingredients induce change in the direct (or volition) target of treatment.

||Ingredients–the observable, measurable and active elements of treatment that are hypothesised to directly change the target of treatment.

**Enhanced assessment inclusive of 1) pain with the Algoplus dementia friendly pain scale [45] at initial and all subsequent therapist contacts; 2) depression (for those with no prior diagnosis) by the therapist using the Cornell Scale for Depression in Dementia [46] after first week in hospital is indicative of a probable or definite depression, then therapist notifies triggers discussion of depression management options with multidisciplinary team; and 3) whether the participant has an advanced care plan. If none, refer to specialty who is responsible for supporting creation of an advanced care plan e.g. general practitioner following discharge.

Secondary objectives of the study will be met by analysis of data indicating potentially eligible participants being screened, eligible, approached, and randomised, consent, and completion CRFs (secondary objective 1), analysis of assessments at intervention end and 12-weeks post-randomisation (secondary objectives 4, 5, 7), and qualitative interviews of patients, carers, and therapists at intervention end (secondary objectives 2, 3, 4). Further secondary objectives are to determine the count of AEs and SAEs identified through reporting procedures in place from the point of randomisation to 12-week follow-up.

A SPIRIT schedule of enrolment, interventions, and assessments is provided in Fig 1. A flow diagram providing a schematic overview of the study is provided in Fig 2.

## Setting

Participants will be recruited within 4 days of admission to hospital with hip fracture at a London teaching hospital.

## Eligibility criteria

Inclusive eligibility criteria will be employed to maximize representativeness of the population (and to ensure that the intervention is assessed for patients in all three risk subgroups). This decision followed PPI consultations and findings from a systematic review detailing inequities in access to trials of rehabilitation after hip fracture surgery (27.3% of potential participants of 35 trials were excluded based on factors that stratify healthcare opportunities and outcomes) [8].

## Patient participant inclusion

➢ aged 60 years or over.

➢ admitted to hospital for surgical repair of hip fracture following low energy trauma.

➢ who are willing to provide consent or by consultee declaration (depending on capacity).

## Patient participant exclusion

➢ less than 60 years, to align with the National Hip Fracture Databases definition of the target population [1].

➢ not surgically treated, as this treatment approach is reserved for around 2% of patients in the UK who are often at the end of life [1].

➢ who broke their hip in hospital following admission for a different illness/injury as their anticipated care pathway and outcomes will vary from those who are admitted for hip fracture.

➢ who broke their hip following a high energy trauma e.g. road traffic accident.

➢ participating in other treatment trials unless both trial Chief Investigators agree to co-enrollment.

➢ who declined to provide consent or by consultee declaration (depending on capacity).

## Professional eligibility

To minimize the potential for contamination, therapists involved in the delivery of the Stratify intervention arm of the feasibility trial will not deliver usual care. Therapists who delivered the

| Data | Form | Source | Completed by | Time point for collection | | | | | |
| --- | --- | --- | --- | --- | --- | --- | --- | --- | --- |
| | | | | Recruitment | Baseline | Randomisation | Intervention (In hospital) | Intervention end | 12-week follow-up |
| Screening log | Binary and Categorical | Patient notes | Site therapist | X | | | | | |
| Subgroup assignment log | Categorical | Patient notes | Site therapist | X | | | | | |
| Approach log | Binary and Categorical | Patient interview | Site therapist | X | | | | | |
| Contact details | Free text | Patient interview | Site therapist | X | | | | | |
| Consent log | Binary | Patient interview | Research team | X | | | | | |
| Age | Numerical | Patient interview | Site therapist | | X | | | | |
| Sex | Binary | Patient interview | Site therapist | | X | | | | |
| Ethnicity | Categorical | Patient interview | Site therapist | | X | | | | |
| Fracture type | Categorical | Patient notes | Site therapist | | X | | | | |
| Surgery type | Categorical | Patient notes | Site therapist | | X | | | | |
| Abbreviated Mental Test | Numerical | Patient notes | Site therapist | | X | | | | |
| Mini Nutritional Assessment | Numerical | Patient notes | Site therapist | | X | | | | |
| Hospital concerns about falls assessment | Numerical | Patient interview | Research team | | X | | | | |
| Residence | Categorical | Patient interview | Site therapist | | X (prefracture) | | | X | X |
| Living status | Categorical | Patient interview | Site therapist | | X (prefracture) | | | X | X |
| Mobility | Categorical | Patient interview | Site therapist | | X (prefracture) | | | | X |
| EuroQoL EQ-5D-5L | Numerical | Patient questionnaire | Research team | | X | | | X | X |
| Barthel Index | Numerical | Patient questionnaire | Research team | | X | | | X | X |
| Nottingham Extended Activities of Daily Living | Numerical | Patient questionnaire | Research team | | X | | | X | X |
| Short Falls Efficacy Scale-International | Numerical | Patient questionnaire | Research team | | X | | | X | X |
| Numeric Rating Scale | Numerical | Patient questionnaire | Research team | | X | | | X | X |
| New Mobility Score | Numerical | Patient questionnaire | Research team | | X | | | X | X |
| Bespoke resource use form | Categorical | Patient questionnaire | Research team | | X | | | | X |
| Randomisation log | Binary | Computer generated randomisation | Research team | | | X | | | |
| Carer screening log | Binary and Categorical | Carer interview | Site therapist | | | | X | | |
| Carer approach log | Binary and Categorical | Carer interview | Site therapist | | | | X | | |
| Carer contact details | Free text | Carer interview | Site therapist | | | | X | | |
| Carer consent log | Binary and Categorical | Carer interview | Research team | | | | X | | |
| Carer age | Numerical | Carer interview | Research team | | | | X | | |
| Carer sex | Binary | Carer interview | Research team | | | | X | | |
| Carer education | Categorical | Carer interview | Research team | | | | X | | |
| Carer children | Binary | Carer interview | Research team | | | | X | | |
| Carer employment status | Categorical | Carer interview | Research team | | | | X | | |
| Carer relationship to participant | Categorical | Carer interview | Research team | | | | X | | |
| Treatment - control | - | - | Site therapist | | | | X | | |
| Treatment - intervention | - | - | Site therapist | | | | X | | |
| Treatment observations | Binary, categorical, and free text | Questionnaire | Research team | | | | X | | |
| Treatment logs | Categorical | Therapist questionnaire | Site therapist | | | | X | | |
| Deviation log | Free text | Therapist questionnaire | Site therapist | | | | X | | |
| Patient diary | Categorical | Patient diary | Patient/carer | | | | X | | |
| Length of stay | Numerical | Patient notes | Site therapist | | | | | X | |
| Mortality | Binary | Online death records | Research team | | | | | X | X |
| Readmission | Binary | Patient interview | Research team | | | | | | X |
| Readmission diagnosis (as applicable) | Free text | Patient interview | Research team | | | | | | X |
| Completion logs | Binary | Therapist questionnaire | Site therapist | | | | | X | X |
| Patient semi-structured interviews | Free text | Patient interview | Research team | | | | | X | |
| Carer semi-structured interviews | Free text | Patient interview | Research team | | | | | X | |
| Therapist semi-structured interviews | Free text | Therapist interview | Research team | | | | | X | |

**Fig 1. Schedule of enrolment, interventions, and assessments.**

Stratify intervention following intervention training (remote 2-hour training session, access to intervention materials, research team e-mail support) will be invited to complete semi-structured interviews focused on treatment acceptability and fidelity (inclusive of barriers and facilitators to implementation).

## Recruitment

Participants will be recruited within 4 days of admission to a hospital ward at St Thomas's Hospital with hip fracture. This is to allow sufficient time to deliver the intervention, given the

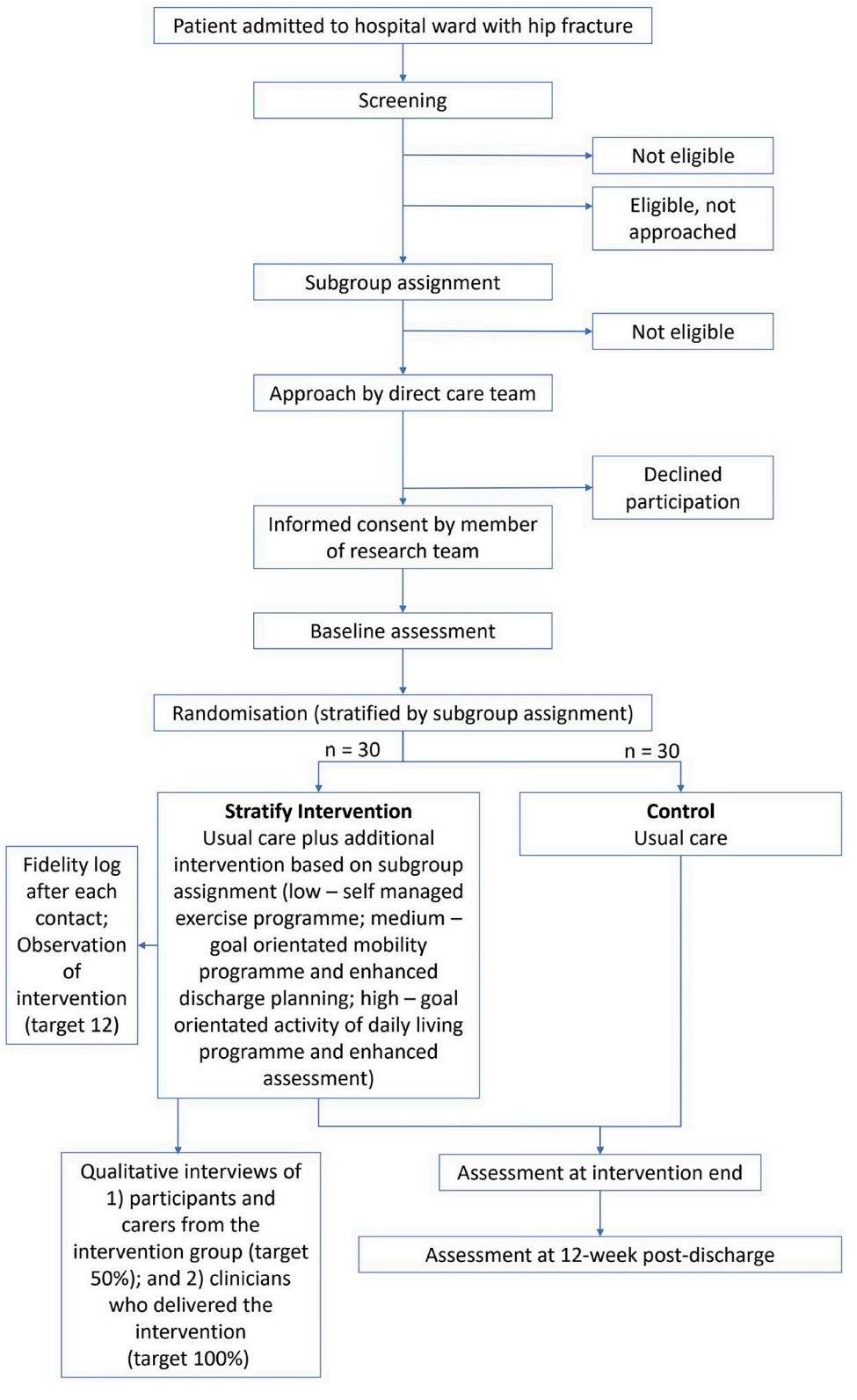

**Fig 2. Schematic overview of the study.**

average length of stay after hip fracture is 16 days [1]. Recruitment of this patient population within this timeframe has been completed successfully in previous UK trials [47]. Patients may be recruited pre- or post- operatively. In the rare instance where a patient is recruited pre-operatively but then does not go on to have surgery, they will be removed from the trial and replaced.

A member of the direct care team will decide whether a potential participant has capacity to give informed consent.

## Participants with capacity to consent

Potential participants will first be approached by a member of the direct care team who will determine their interest in taking part in the trial and gain consent to share their contact details with a member of the research team. Consent to contact will be documented in a CRF and medical notes. A member of the research team will subsequently provide an explanation of the aims, methods, benefits and potential harms and a participation information leaflet. Potential participants will be given at least 24 hours to consider their participation in the study. A member of the research team will subsequently answer any questions prior to obtaining written informed consent to enter and be randomised into the trial. Non-English language speaking older adults will be supported to consider their enrolment in the trial with the use of language support at the hospital.

## Participants without capacity to consent

For those who are considered as lacking capacity, agreement will be sought by a member of the research team from the patient and their consultee, in accordance with the approach recommended by the Mental Capacity Act, 2005 [48]. If agreement is provided, the consultee will be provided with a consultee participant information leaflet and asked to sign a consultee declaration form. If the patient does not have a next of kin, the consultee may be a member of the patient's direct care team (who is not also a member of the research team). Subsequently, the member of the research team will seek assent from the patient for participation, as able. If a participant who lacks capacity indicates dissent, but their consultee advises involvement in the study, they will not be included in the study. Consultees will be invited to support the participant with data collection, or complete data collection on a participant's behalf. The number of consultees who declined to support data collection (and why) will be documented.

It is possible that a participant without capacity may see a capacity improvement during the study (e.g., for participants with pre/postoperative delirium that resolves). If this occurs, they will be informed of their enrolment in the study and what it involves, including the provision of the participant information leaflet and the opportunity to consent for themselves. They will be advised that they are free to withdraw from the trial at any point. It is unlikely but possible a participant with capacity may see a capacity deterioration during the study. If this occurs, their original consent (with capacity) will be preserved. A consultee will be sought, provided a consultee participant information leaflet, and asked to sign a consultee declaration form. The consultee will be invited to support the participant with data collection, or complete data collection on a participant's behalf.

## Carers

For all participants (irrespective of capacity status) enrolled in the intervention arm of the medium- or high-risk subgroups, their carer will be invited to join the intervention sessions, if their carer is willing and able to provide consent. A member of the direct care team will approach their carer in person or over the phone to tell them about the study and decide

whether the carer has capacity to give informed consent. If so, the clinician will determine their interest in taking part and gain consent to share their contact details with a member of the research team. Screening, eligibility, approach, and consent to contact will be documented in the CRF. A member of the research team will subsequently provide an explanation of the aims, methods, benefits and potential harms and a carer information leaflet. Carers will be given at least 24 hours to consider their participation. A member of the research team will subsequently answer any questions prior to obtaining written informed consent. The number of carers who declined (and why) will be documented. Should a carer decline to take part, this will not affect the patient participant enrolled in the study.

## Therapists

A member of the research team will outline the aims, methods, benefits and potential harms of the qualitative study and provide a participation information leaflet during therapist training prior to the start of the study. Therapists will be asked to provide consent to contact from the research team at this training (their name and email address will be stored on an Excel Spreadsheet on a secure server). At the end of the study a member of the research team will contact those who provided consent to contact and answer any questions prior to obtaining written informed consent to the interview study.

## Randomization and allocation procedure

The randomization approach was designed with a statistician (SA) and validated for use prior to implementation. Once a participant has provided informed consent, they will be allocated a participant identification number. Baseline data will subsequently be collected by the research team prior to randomization. Randomization will follow a 1:1 random allocation sequence stratified by subgroup assignment (low-, medium-, high- risk), using a secure internet-based system, developed, and maintained by King's Clinical Trials Unit to ensure allocation concealment. Treatment allocation will be revealed (after randomization) to a member of the research team, linked to the participant identification number, and the clinical team will be notified.

## Sample size

The recruitment target aims to have sufficient participants to provide the operational experience to plan a definitive future trial; provide reasonably robust estimates of our feasibility outcomes; and to estimate the variability of the proposed patient outcomes to inform a future sample size calculation. A recruitment target of 60 participants (30 per treatment arm) will allow overall retention rate at 12-weeks to be estimated with precision of ±11%, using an exact 95% confidence interval, from previously observed retention rates of ~80% for the same population [49]. Assuming a non-differential retention rate of 80% at 12-week follow-up, this target will provide follow-up outcome data on ~24 participants per arm.

## Data collection and outcomes

Participants will undergo screening, baseline assessment, assessment at intervention end, and assessment at 12-weeks post-randomisation. Assessments at baseline and intervention end will be in-person, over the telephone, or via MS TEAMS (the mode will be documented). Assessments at 12-weeks post-randomisation will be completed over the telephone or via MS TEAMS. Outcome assessors will be blind to group allocation. Participants who are non-English language speakers will be supported to complete assessments (except patient reported outcome measures) with the use of language support at the hospital. Patient-reported outcome

measures which have an established translated, validated, and (where applicable) culturally adapted version in the appropriate language will be circulated by post to the participant with a pre-paid envelope for return direct to the research team.

### Baseline

Following consent and prior to randomisation, a member of the research team will collect the following participant characteristics: age, sex, ethnicity, fracture type, surgery type, Abbreviated Mental Test Score, Mini Nutritional Assessment, hospital concerns about falls assessment, pre-fracture residential status (home, residential home, nursing home), living status (lives alone, with independent spouse, with dependent spouse, with family, with other), prefracture mobility. They will collect the following patient-reported outcome measures which satisfy the core outcome set for hip fracture trials [50]:

1. health-related quality of life (EuroQoL EQ-5D-5L [51])

2. activities of daily living (Barthel Index [52]; Nottingham Extended Activities of Daily Living [53])

3. falls related self-efficacy (Short Falls Efficacy Scale-International [54, 55])

4. pain (Numeric Rating Scale [56])

5. walking ability (New Mobility Score [57])

A bespoke resource use data collection form will also be collected. This form can be collected both prior to and/or after randomisation but prior to discharge.

### During the intervention

For carers recruited to join the intervention sessions in the medium- or high- risk groups of the STRATIFY arm of the trial, we will collect data relating to their age, sex, education, employment (not employed, part-time, full time), children (yes, no), and relationship to the participant.

The research team will conduct fidelity observations of at least twelve therapist-led sessions (four assessments for each subgroup in the intervention arm) following verbal consent from the therapist and participant engaged in each session. The research team will conduct fidelity observations of at least four carer training sessions (two assessments for medium- and high-risk subgroup in the intervention arm). Participants in the intervention arm will be asked to complete a participant diary indicating the extent to which they practiced training without healthcare professionals.

### Intervention end

At the intervention end, a member of the research team will collect data on hospital length of stay, mortality (death records will be checked online prior to contacting the participant/their carer), place of residence, living status, adverse/serious adverse events, as well as the patient-reported outcome measures.

Participants with different risk subgroup assignment (low, medium, high) and carers of those assigned to medium- and high-risk subgroups will be purposively sampled for telephone/MS TEAMS semi-structured interviews focused on acceptability (of subgroup assignment and matched treatment) and fidelity (training, delivery, receipt and enactment). Interviews will target 50% of intervention participants and 50% of carers but continue until no new themes are identified [58].

Therapists involved in the delivery of the STRATIFY intervention arm will be followed up by the research team by email to seek informed consent to participate in telephone/MS TEAMS semi-structured interviews focused on treatment acceptability (of subgroup assignment and matched treatment) and fidelity (training, delivery, and enactment). Given the proposed study is single site, interviews will target 100% of therapist involved in delivery of the intervention arm.

## 12-week follow-up

At 12-week follow up, a member of the research team will collect data related to mortality, readmissions (and admitting diagnosis), place of residence, and living status, any adverse/serious adverse events, as well as the PROMs. The bespoke resource use data collection form will also be collected at this follow-up.

## Adverse events

Participant safety will be determined through the reporting of adverse events (AE) and serious adverse events (SAE). AEs to be collected and reported will include an exacerbation of a pre-existing illness, an increase in the frequency or intensity of a pre-existing episodic event or condition, and/or continuous persistent disease or a symptom present at baseline that worsens following administration of the trial intervention. A SAE is an untoward occurrence that results in death, is life threatening (at the time of the event), requires unplanned hospitalization or prolongation of an existing hospitalization, and/or results in persistent or significant disability or incapacity. Other 'important medical events' may be considered serious if they jeopardize the participant or require an intervention to prevent one of the above SAEs. Where any AE/SAE occur, the team will adhere to guidelines for the reporting to the medical team, Trial Oversight Committee, and the research team who will subsequently assess relatedness to the intervention and report to the relevant sponsors and regulators.

## Data analysis

A statistical analysis plan will be finalised ahead of database locking and reporting will follow the CONSORT guidance for pilot and feasibility studies [59]. All participants who are randomised will be included in analyses according to the group they were originally assigned, regardless of treatment received.

## Quantitative

The statistician will be blind to group allocation. A CONSORT flow diagram will display data specifying counts of screened, eligible, approached, randomised, and completed enabling estimation of eligibility, recruitment, consent and follow-up rates [59]. Confidence intervals for recruitment and retention rates will be produced to inform assumptions for planning the definitive trial. Rates of eligible, approach and recruited will also be estimated for carers of participants enrolled into the medium- or high-risk intervention arm. Completion rates will be estimated for outcome measures collected at each time-point. Baseline characteristics will be summarised by allocated arm (and by subgroup assignment) with descriptive statistics (measures of central tendency and dispersion) to enable assessment of baseline comparability of arms (a degree of imbalance is anticipated in this feasibility and pilot trial). Patient-reported outcomes and treatment fidelity (inclusive of therapist time) will be summarised by allocated arm (and by subgroup assignment) at each follow-up, with descriptive statistics (measures of central tendency and dispersion). Between-arm differences, including changes from baseline,

will be reported for the PROMs with corresponding measures of dispersion to enable an assessment of sensitivity to change to inform primary outcome selection for a definitive trial and a formal power calculation for this outcome for a definitive trial. Rate and proportion of missing data will be reported for all analyses, with reasons where known.

For patients who do not speak English, the count of pseudo-anonymised patient-reported outcome measures circulated and returned by post will be documented. The content of the PROM will be described narratively ensuring participant anonymity is preserved (the language versions circulated will not be specified in reporting).

The analysis will be completed by a trial statistician using R (https://www.r-project.org/) after database lock at trial end. A Trial Oversight Committee will monitor screening/eligibility, approach and randomisation rates and safety reporting. There are no additional interim analyses planned.

## Qualitative

Qualitative data will be transcribed verbatim from semi-structured interviews and analysed using a thematic analysis approach [60]. The analysis will follow a deductive approach informed by the categories of treatment fidelity (design, training of providers, delivery, receipt and enactment), and to identify barriers and/or facilitators (inclusive of acceptability and aligned to Normalisation Process Theory [14]) to future implementation [61]. The Stratify intervention observations will be sampled against therapists CRFs to further assess fidelity. Free text entries made during observations will be summarised narratively.

## Progression criteria

To mitigate the risk of ongoing uncertainty at the end of the feasibility and pilot trial, we propose progression criteria outlined in Fig 3 [62].

## Monitoring

The trial will be conducted in compliance with the approved protocol, to Good Clinical Practice, the UK General Data Protection Regulation and Data Protection Act (2018), the local Information Governance Policy, the UK Policy Framework for Health and Social Care Research, the sponsor's Standard Operating Procedures, the Mental Capacity Act 2005, and other applicable regulations as required. The Trial Management Group will establish a Monitoring Plan inclusive of data monitoring for accuracy and completeness, periodic review of adverse events, critical data monitoring (including subgroup assignment), and eligibility prior to randomization. A Trial Oversight Committee comprised of members of the research team and independent members (chair, statistician, expert members, public representatives) will provide advice, data monitoring (screening and recruitment rates, accruing outcome data), quality assurance, and safety monitoring (number, nature and outcomes for all serious adverse events). The committee may include open and closed sessions. Closed sessions will not be attended by blinded members of the research team and may be used for data monitoring and/ or other discussions at the discretion of the chair. The committee will be asked to recommend any necessary actions. It is anticipated that the committee will meet at least biannually during the trial period.

## Public and patient involvement

Patients and carers were involved from proposal conception onwards. A 2-hour focus group of patients and carers informed the 1) target population as many carers expressed concern over

| | GO | AMEND | STOP |
|---|---|---|---|
| **Recruitment** | ≥40% eligible | 21-39% eligible | ≤20% eligible |
| **Recruitment** | ≥50% eligible recruited | 31-49% eligible recruited | ≤30% eligible recruited |
| **Randomisation** | ≥70% of those recruited randomised | 49-70% of those recruited randomised | ≤48% of those recruited randomised |
| **Fidelity** | ≥80% sessions included all intervention components as described | 51-79% sessions included all intervention components as described | ≤50% sessions included all intervention components as described |
| **Outcome, 12-weeks** | ≥80% completeness of EQ5D at 12-week follow-up | 51-79% completeness of EQ5D at 12-week follow-up | ≤50% completeness of EQ5D at 12-week follow-up |

**Fig 3. Progression criteria.**

disparities in the provision of rehabilitation based on patient characteristics such as the presence of dementia and/or admission from a residential/nursing care home (population selected–all patients surgically treated for hip fracture), and 2) outcomes for the subgrouping algorithm as all who took part expressed a goal of returning home as soon as possible (outcomes selected–time to death and time to change in residence).

In 2020, a PPI group was established which meets quarterly to discuss Trauma Rehabilitation (Orthopaedic) research for Older People—'TROOP'. TROOP includes men and women from different ethnic backgrounds who reside at home across England. Members of TROOP contributed to the:

- intervention design during a workshop 06/22 and focus group 01/23. Members were given the opportunity to provide suggestions for the intervention and feedback on suggestions from others during a workshop and a focus group.

- development of educational materials to be provided as part of the intervention (focus group 04/23 and subsequent written feedback on materials via email).

- draft of the participant information leaflet and consent forms (written feedback via email).

This active collaboration with PPI members (currently eight members) of TROOP will continue for the duration of the project. The UK Standards for Public Involvement will be followed to ensure this collaboration follows best practice [63]. AFM agreed to take a leadership role for TROOP with respect to the current project and attend trial management group meetings. She will be sent materials (including an outline of the format) in advance of each meeting.

She will be offered a pre-meeting with the lead applicant to discuss anything that is not clear. TROOP members will also continue to meet quarterly. These meetings will include discussion of progress, interpretation of qualitative and quantitative results, and development and dissemination of plain English summaries of the project findings. Finally, two PPI members independent of TROOP were recruited for the Trial Oversight Committee through NIHR People in Research.

## Dissemination

The results of the feasibility of a future trial will be summarised in plain English and made available on the teams public and patient involvement group webpage (www.ppitroop.co.uk) and Twitter page (@TROOP_PPI) as well as via the Royal Osteoporosis Society's Bone Matters e-newsletter (circulations in excess of 20,000). Participants will be offered the option of having the plain English summary posted directly to them during the consent process. If any non-English language speakers take part in the trial, the summary will be translated into the appropriate language prior to posting directly to them.

The results of the study will be published in open-access peer reviewed journals. The findings will be presented at national conferences (British Geriatrics Society; British Orthopaedic Association) and international conferences (Fragility Fracture Network (FFN)).

Results will also be disseminated through the European Geriatric Medicine Society (past president Martin), Chartered Society of Physiotherapy (members Sheehan, Sackley, Gibson, Jones); Royal College of Occupational Therapists (Fellow, Sackley); and the FFN (Chair-elect Scientific Committee, Sheehan).

Following publication of the primary paper, an anonymised dataset will be preserved indefinitely on the King's Open Research Data System with proof of ethical approval as a condition of access (https://www.kcl.ac.uk/researchsupport/managing/preserve).

## Supporting information

**S1 File. SPIRIT checklist.**
(DOC)

**S2 File. Stratify protocol.**
(PDF)

**S3 File. Consent form.**
(PDF)

**S4 File. Carer consent form.**
(PDF)

**S5 File. Consultee declaration form.**
(PDF)

**S6 File. Remote workshop e-booklet.**
(PDF)

**S7 File.**
(PDF)

**S1 Table.**
(ZIP)

## Acknowledgments

We are grateful to the input from patient and public members of the involvement group TROOP (https://www.ppitroop.co.uk/) for their support in the design of this trial, to the participants in the qualitative interview studies and intervention development workshops which underpin the intervention.

## Author Contributions

**Conceptualization:** Katie J. Sheehan, Stefanny Guerra, Salma Ayis, Aicha Goubar, Nadine E. Foster, Finbarr C. Martin, Emma Godfrey, Ian D. Cameron, Celia L. Gregson, Nicola E. Walsh, Anna Ferguson Montague, Rebecca Edwards, Jodie Adams, Gareth D. Jones, Jamie Gibson, Catherine Sackley, Julie Whitney.

**Funding acquisition:** Katie J. Sheehan.

**Methodology:** Katie J. Sheehan, Stefanny Guerra, Salma Ayis, Aicha Goubar, Nadine E. Foster, Finbarr C. Martin, Emma Godfrey, Ian D. Cameron, Celia L. Gregson, Nicola E. Walsh, Anna Ferguson Montague, Rebecca Edwards, Jodie Adams, Gareth D. Jones, Jamie Gibson, Catherine Sackley, Julie Whitney.

**Writing – original draft:** Katie J. Sheehan, Stefanny Guerra.

**Writing – review & editing:** Katie J. Sheehan, Stefanny Guerra, Salma Ayis, Aicha Goubar, Nadine E. Foster, Finbarr C. Martin, Emma Godfrey, Ian D. Cameron, Celia L. Gregson, Nicola E. Walsh, Anna Ferguson Montague, Rebecca Edwards, Jodie Adams, Gareth D. Jones, Jamie Gibson, Catherine Sackley, Julie Whitney.

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
