## [Decision Letter · Decision Letter 0]

2 May 2024

PONE-D-24-12423Structured Tailored Rehabilitation after Hip Fragility Fracture: The ‘Stratify’ Feasibility and Pilot Randomised Controlled Trial ProtocolPLOS ONE

Dear Dr. Sheehan,

Thank you for submitting your manuscript to PLOS ONE. After careful consideration, we feel that it has merit but does not fully meet PLOS ONE’s publication criteria as it currently stands. Therefore, we invite you to submit a revised version of the manuscript that addresses the points raised during the review process.

We look forward to receiving your revised manuscript.

Kind regards,

Sina Azadnajafabad, MD, MPH

Academic Editor

PLOS ONE

Journal Requirements:

2. Please expand the acronym “UKRI” (as indicated in your financial disclosure) so that it states the name of your funders in full.

"We are grateful to the input from patient and public members of the involvement group TROOP(https://www.ppitroop.co.uk/) for their support in the design of this trial, to the participants in the qualitative interview studies and intervention development workshops which underpin the intervention. This work acknowledges the support of the National Institute for Health Research Barts Biomedical Research Centre (NIHR203330)."

"This paper presents independent research funded by UKRI Future Leaders Fellowship [Grant Ref: MR/S032819/1] awarded to KS. The funders had no role in study design, data collection and analysis, decision to publish, or preparation of the manuscript. Sponsors URL: https://www.ukri.org/what-we-do/developing-people-and-skills/future-leaders-fellowships/."

"I have read the journal's policy and the authors of this manuscript have the following competing interests: KS received a grant from UK Research & Innovation Future Leaders Fellowship to support this work. This funding provides salary support for KS. KS also received funding from the National Institutes of Health Research (NIHR) and the Royal Osteoporosis Society for hip fracture health services research. KS is the Chair and CG a member of the Scientific and Publications Committee of the Falls and Fragility Fracture Audit Programme which managed the National Hip Fracture Database audit at the Royal College of Physicians. FCM was the funded (2012-2018) board chair of the Falls and Fragility Fracture programme. NEF is funded through an Australian National Health and Medical Research Council (NHMRC) Investigator Grant (ID: 2018182). CS and NW receive funding from the National Institute for Health Research (NIHR). CS is an NIHR Senior Investigator. Salma Ayis was funded/supported by the National Institute for Health Research (NIHR) Biomedical Research Centre based at Guy's and St Thomas' NHS Foundation Trust and King's College London. The views expressed are those of the author(s) and not necessarily those of the NHS, the NIHR or the Department of Health. CLG receives funding from Versus Arthritis (ref 22086). GSdP, SA, and IDC have no competing interests to declare."

8. We note that the original protocol file you uploaded contains a confidentiality notice indicating that the protocol may not be shared publicly or be published. Please note, however, that the PLOS Editorial Policy requires that the original protocol be published alongside your manuscript in the event of acceptance. Please note that should your paper be accepted, all content including the protocol will be published under the Creative Commons Attribution (CC BY) 4.0 license, which means that it will be freely available online, and any third party is permitted to access, download, copy, distribute, and use these materials in any way, even commercially, with proper attribution.

Therefore, we ask that you please seek permission from the study sponsor or body imposing the restriction on sharing this document to publish this protocol under CC BY 4.0 if your work is accepted. We kindly ask that you upload a formal statement signed by an institutional representative clarifying whether you will be able to comply with this policy. Additionally, please upload a clean copy of the protocol with the confidentiality notice (and any copyrighted institutional logos or signatures) removed.

9. We note that the original protocol that you have uploaded as a Supporting Information file contains an institutional logo. As this logo is likely copyrighted, we ask that you please remove it from this file and upload an updated version upon resubmission.

**Additional Editor Comments:**

Invited reviewers has raised some comments which need to be addressed before any final decision.

Reviewers' comments:

Reviewer's Responses to Questions

**Comments to the Author**

1. Does the manuscript provide a valid rationale for the proposed study, with clearly identified and justified research questions?

Reviewer #1: Yes

Reviewer #2: Yes

2. Is the protocol technically sound and planned in a manner that will lead to a meaningful outcome and allow testing the stated hypotheses?

Reviewer #1: Yes

Reviewer #2: Yes

3. Is the methodology feasible and described in sufficient detail to allow the work to be replicable?

Reviewer #1: Yes

Reviewer #2: Yes

4. Have the authors described where all data underlying the findings will be made available when the study is complete?

Reviewer #1: Yes

Reviewer #2: Yes

5. Is the manuscript presented in an intelligible fashion and written in standard English?

Reviewer #1: Yes

Reviewer #2: Yes

6. Review Comments to the Author

You may also provide optional suggestions and comments to authors that they might find helpful in planning their study.

Reviewer #1: The 'Stratify' Feasibility and Pilot Randomised Controlled Trial Protocol appears to be well-designed with a comprehensive approach to ensuring compliance with regulations, monitoring, and patient involvement together with the detailed intervention specifications for different risk subgroups show a thoughtful approach to tailoring treatments.

Just a minor point for consideration

Is there risk of the participants being incorrectly assigned to a subgroup that does not align with their actual risk level which could lead to suboptimal treatment allocation and potentially impact the study outcomes.

In the analysis section, can researchers mention briefly approach of missing data.

Reviewer #2: Dear authors,

Thank you for the effort you've put into this study, and I'm also grateful for the opportunity to review this manuscript. I believe that this study has the potential to serve as a foundation for larger-scale studies on the topic of hip fracture tailored rehabilitation.

7. PLOS authors have the option to publish the peer review history of their article (what does this mean?). If published, this will include your full peer review and any attached files.

Reviewer #1: No

Reviewer #2: **Yes: **Sina Afzal

---

## [Author Response · Author response to Decision Letter 0]

30 May 2024

Editors comments

a. Author response: completed.

2. Please expand the acronym “UKRI” (as indicated in your financial disclosure) so that it states the name of your funders in full. This information should be included in your cover letter; we will change the online submission form on your behalf.

a. Author response: United Kingdom Research and Innovation (UKRI)

3. Thank you for stating the following in the Acknowledgments Section of your manuscript: "We are grateful to the input from patient and public members of the involvement group TROOP(https://www.ppitroop.co.uk/) for their support in the design of this trial, to the participants in the qualitative interview studies and intervention development workshops which underpin the intervention. This work acknowledges the support of the National Institute for Health Research Barts Biomedical Research Centre (NIHR203330)."We note that you have provided funding information that is not currently declared in your Funding Statement. However, funding information should not appear in the Acknowledgments section or other areas of your manuscript. We will only publish funding information present in the Funding Statement section of the online submission form. Please remove any funding-related text from the manuscript and let us know how you would like to update your Funding Statement. Currently, your Funding Statement reads as follows: "This paper presents independent research funded by UKRI Future Leaders Fellowship [Grant Ref: MR/S032819/1] awarded to KS. The funders had no role in study design, data collection and analysis, decision to publish, or preparation of the manuscript. Sponsors URL: https://www.ukri.org/what-we-do/developing-people-and-skills/future-leaders-fellowships/."Please include your amended statements within your cover letter; we will change the online submission form on your behalf.

a. Author response: We have removed the text from the acknowledgement section. Please update the funding statement to read: ‘This work was supported by a UKRI Future Leaders Fellowship [Grant Ref: MR/S032819/1]. This work acknowledges the support of the National Institute for Health Research Barts Biomedical Research Centre (NIHR203330).’

4. Thank you for stating the following in the Competing Interests section: "I have read the journal's policy and the authors of this manuscript have the following competing interests: KS received a grant from UK Research & Innovation Future Leaders Fellowship to support this work. This funding provides salary support for KS. KS also received funding from the National Institutes of Health Research (NIHR) and the Royal Osteoporosis Society for hip fracture health services research. KS is the Chair and CG a member of the Scientific and Publications Committee of the Falls and Fragility Fracture Audit Programme which managed the National Hip Fracture Database audit at the Royal College of Physicians. FCM was the funded (2012-2018) board chair of the Falls and Fragility Fracture programme. NEF is funded through an Australian National Health and Medical Research Council (NHMRC) Investigator Grant (ID: 2018182). CS and NW receive funding from the National Institute for Health Research (NIHR). CS is an NIHR Senior Investigator. Salma Ayis was funded/supported by the National Institute for Health Research (NIHR) Biomedical Research Centre based at Guy's and St Thomas' NHS Foundation Trust and King's College London. The views expressed are those of the author(s) and not necessarily those of the NHS, the NIHR or the Department of Health. CLG receives funding from Versus Arthritis (ref 22086). GSdP, SA, and IDC have no competing interests to declare." Please confirm that this does not alter your adherence to all PLOS ONE policies on sharing data and materials, by including the following statement: "This does not alter our adherence to PLOS ONE policies on sharing data and materials.” (as detailed online in our guide for authors http://journals.plos.org/plosone/s/competing-interests). If there are restrictions on sharing of data and/or materials, please state these. Please note that we cannot proceed with consideration of your article until this information has been declared. Please include your updated Competing Interests statement in your cover letter; we will change the online submission form on your behalf.

a. Author response: Please update to read: “I have read the journal's policy and the authors of this manuscript have the following competing interests: KS received a grant from UK Research & Innovation Future Leaders Fellowship to support this work. This funding provides salary support for KS. KS also received funding from the National Institutes of Health Research (NIHR) and the Royal Osteoporosis Society for hip fracture health services research. KS is the Chair and CG a member of the Scientific and Publications Committee of the Falls and Fragility Fracture Audit Programme which managed the National Hip Fracture Database audit at the Royal College of Physicians. FCM was the funded (2012-2018) board chair of the Falls and Fragility Fracture programme. NEF is funded through an Australian National Health and Medical Research Council (NHMRC) Investigator Grant (ID: 2018182). CS and NW receive funding from the National Institute for Health Research (NIHR). CS is an NIHR Senior Investigator. Salma Ayis was funded/supported by the National Institute for Health Research (NIHR) Biomedical Research Centre based at Guy's and St Thomas' NHS Foundation Trust and King's College London. The views expressed are those of the author(s) and not necessarily those of the NHS, the NIHR or the Department of Health. CLG receives funding from Versus Arthritis (ref 22086). GSdP, SA, and IDC have no competing interests to declare. This does not alter our adherence to PLOS ONE policies on sharing data and materials.”

a. Author response: Completed

a. Author response: We removed the ethics statement from the abstract. It now only appears in the methods section of the manuscript. 

a. Author response: Completed

8. We note that the original protocol file you uploaded contains a confidentiality notice indicating that the protocol may not be shared publicly or be published. Please note, however, that the PLOS Editorial Policy requires that the original protocol be published alongside your manuscript in the event of acceptance. Please note that should your paper be accepted, all content including the protocol will be published under the Creative Commons Attribution (CC BY) 4.0 license, which means that it will be freely available online, and any third party is permitted to access, download, copy, distribute, and use these materials in any way, even commercially, with proper attribution. Therefore, we ask that you please seek permission from the study sponsor or body imposing the restriction on sharing this document to publish this protocol under CC BY 4.0 if your work is accepted. We kindly ask that you upload a formal statement signed by an institutional representative clarifying whether you will be able to comply with this policy. Additionally, please upload a clean copy of the protocol with the confidentiality notice (and any copyrighted institutional logos or signatures) removed.

a. Author response: We cannot locate any confidentiality notice in the original protocol file uploaded. No changes have been made with respect to this. The logos and signatures have been removed. 

9. We note that the original protocol that you have uploaded as a Supporting Information file contains an institutional logo. As this logo is likely copyrighted, we ask that you please remove it from this file and upload an updated version upon resubmission.

a. Author response: Logo removed.

a. Author response: No changes to the reference list. 

Reviewers comments

Reviewer #1: 

1. The 'Stratify' Feasibility and Pilot Randomised Controlled Trial Protocol appears to be well-designed with a comprehensive approach to ensuring compliance with regulations, monitoring, and patient involvement together with the detailed intervention specifications for different risk subgroups show a thoughtful approach to tailoring treatments. Just a minor point for consideration, Is there risk of the participants being incorrectly assigned to a subgroup that does not align with their actual risk level which could lead to suboptimal treatment allocation and potentially impact the study outcomes.

a. Author response: We updated the ‘monitoring’ section to read ‘The Trial Management Group will establish a Monitoring Plan inclusive of data monitoring for accuracy and completeness, periodic review of adverse events, critical data monitoring (including subgroup assignment), and eligibility prior to randomization.’ We also updated the section ‘Data collection and outcomes, intervention end’ to read: ‘….interviews focused on acceptability (of subgroup assignment and matched treatment) and fidelity (training, delivery, receipt and enactment)’ to clarify the plan of how we will meet two key objectives of this feasibility trial.

2. In the analysis section, can researchers mention briefly approach of missing data.

a. Author response: We updated the ‘data analysis’ section to read: ‘All participants who are randomised will be included in analyses according to the group they were originally assigned, regardless of treatment received.’ and ‘Rate and proportion of missing data will be reported for all analyses with reasons where known’.

Reviewer #2: 

1. Dear authors, Thank you for the effort you've put into this study, and I'm also grateful for the opportunity to review this manuscript. I believe that this study has the potential to serve as a foundation for larger-scale studies on the topic of hip fracture tailored rehabilitation.

a. Author response: Many thanks for taking the time to review the protocol for the feasibility trial.

---

## [Decision Letter · Decision Letter 1]

25 Jun 2024

Structured Tailored Rehabilitation after Hip Fragility Fracture: The ‘Stratify’ Feasibility and Pilot Randomised Controlled Trial Protocol

PONE-D-24-12423R1

Dear Dr. Sheehan,

We’re pleased to inform you that your manuscript has been judged scientifically suitable for publication and will be formally accepted for publication once it meets all outstanding technical requirements.

Kind regards,

Sina Azadnajafabad, MD, MPH

Academic Editor

PLOS ONE

Additional Editor Comments (optional):

Reviewers' comments:

Reviewer's Responses to Questions

**Comments to the Author**

1. Does the manuscript provide a valid rationale for the proposed study, with clearly identified and justified research questions?

Reviewer #1: Yes

Reviewer #2: Yes

2. Is the protocol technically sound and planned in a manner that will lead to a meaningful outcome and allow testing the stated hypotheses?

Reviewer #1: Yes

Reviewer #2: Yes

3. Is the methodology feasible and described in sufficient detail to allow the work to be replicable?

Reviewer #1: Yes

Reviewer #2: Yes

4. Have the authors described where all data underlying the findings will be made available when the study is complete?

Reviewer #1: Yes

Reviewer #2: Yes

5. Is the manuscript presented in an intelligible fashion and written in standard English?

Reviewer #1: Yes

Reviewer #2: Yes

6. Review Comments to the Author

You may also provide optional suggestions and comments to authors that they might find helpful in planning their study.

Reviewer #1: All comments have been addressed.

Reviewer #2: Thanks for the amendments you've done into the revised version of your manuscript, based on the reviewers' and editor's comments.

7. PLOS authors have the option to publish the peer review history of their article (what does this mean?). If published, this will include your full peer review and any attached files.

Reviewer #1: No

Reviewer #2: **Yes: **Sina Afzal

---

## [Editor Report · Acceptance letter]

9 Aug 2024

PONE-D-24-12423R1 

PLOS ONE

Dear Dr. Sheehan, 

I'm pleased to inform you that your manuscript has been deemed suitable for publication in PLOS ONE. Congratulations! Your manuscript is now being handed over to our production team.

Kind regards, 

on behalf of

Dr. Sina Azadnajafabad 

Academic Editor

PLOS ONE